## PERSPECTIVE

# Bridging the communication gap: cardiomyocytes reciprocate sympathetic nerve signalling

Rebecca J. Salamon 🔘
and Ahmed I. Mahmoud 🔘

*Department of Cell and Regenerative Biology, University of Wisconsin-Madison School of Medicine and Public Health, Madison, WI, USA*

Email: aimahmoud@wisc.edu

Edited by: Bjorn Knollmann & Jian Shi

Linked articles: This Perspective article highlights an article by Dokshokova et al. To read this paper, visit https://doi.org/10.1113/JP282828

The peer review history is available in the Supporting information section of this article (https://doi.org/10.1113/JP283173#support-information-section).

Sympathetic nerves (SNs) innervate the mammalian heart and regulate cardiac function by increasing contraction and rhythm. The precise spatial patterning of cardiac sympathetic innervation during development is the result of tightly regulated events including neurotrophic and neuro-repellant factors from cardiomyocytes (CMs) (Ieda et al., 2007). Importantly, the distinct innervation patterns of SNs allow for precise anterograde neuro-cardiac 'SNs to CMs' communication via the synaptic release of noradrenaline that governs cardiac function at the neuro-cardiac junction (NCJ), which is analogous in name and structure to the neuro-muscular junction (Prando et al., 2018). Interestingly, several cardiovascular diseases are accompanied with secondary myocardial sympathetic denervation, providing evidence that maintenance of SN innervation is dependent on CMs (Gardner et al., 2016). Collectively, these findings demonstrate a significant link between CMs and SNs during homeostasis, and that disruption of this signalling can exacerbate cardiovascular disease. However, whether retrograde signalling from CMs to SNs at the NCJ is required to maintain adult heart SN innervation under homeostasis was unknown. Defining the mechanisms by which CMs and SNs communicate can provide novel avenues to maintain physiological innervation in cardiovascular diseases.

In this issue of *The Journal of Physiology*, Dokshokova and colleagues explore whether a reverse CM to SN communication at the NCJ is required for maintaining adult heart sympathetic innervation and function (Dokshokova et al., 2022). Immunofluorescence staining of rat ventricular sections showed the expression of nerve growth factor (NGF) vesicles in CMs at the NCJ, NGF being the main neurotrophin required for SN innervation and maturation (Clegg et al., 1989). Furthermore, the high-affinity NGF receptor, TrkA, was present on SN neuronal processes. Interestingly, this expression pattern was also detected in myocardial samples from human hearts as well. These findings supported the hypothesis that retrograde communication from CMs to SNs takes place via NGF at the NCJ in the mammalian heart.

To further establish this retrograde communication, the authors performed co-culture experiments with SNs and CMs. Typically, SN survival and maturation in culture requires the addition of exogenous NGF; however, the co-culture of SNs with CMs alone was sufficient to promote survival and maturation of SNs without exogenous NGF. These results demonstrate that CMs provide NGF for SN survival and maturation. To further define whether cell-specific interactions between CMs and SNs mediate this signalling, the authors co-cultured SNs with cardiac fibroblasts (CFs), which synthesize high levels of NGF, but lack an NCJ. Remarkably, the co-culture of SNs with CFs demonstrated significantly lower levels of NGF in the SN processes in contact with CFs in comparison to the CM co-culture. These results reveal that CMs are the source of NGF for the innervating SNs, and this retrograde signalling occurs specifically at the NCJ.

Since sympathetic denervation occurs following CM injury, the authors wanted to determine whether specific inhibition of NGF in CMs can induce degeneration of SNs. The authors treated CMs with an siRNA against NGF, which reduced NGF expression without disrupting the expression of other neurotrophins or impacting CM viability. Interestingly, NGF knockdown in CMs resulted in fragmentation and reduction of the innervating SNs, which were restored by exogenous addition of NGF. To explore if the NGF release from CMs is diffuse or localized at the NCJ, the authors measured NGF levels in the conditioned medium from CMs and quantified very low levels of NGF. In addition, treatment of SN and CM co-culture with a TrkA receptor antagonist reduced neuronal viability. Together, these results demonstrate that specific retrograde release of NGF from CMs is required to maintain survival of the innervating SNs by binding to the TrkA receptor at the NCJ.

Collectively, the findings of Dokshokova et al. demonstrate that CMs and SNs participate in bidirectional cellular communication, specifically via NGF release at the NCJ. This study provides mechanistic evidence relating to the sympathetic denervation that takes place in the context of cardiovascular diseases. The identification of the bidirectional communication between CMs and SNs provides new avenues to maintaining physiological innervation that can ameliorate the extent of cardiac diseases. The precise nature of nerve topology and intracellular connections highlights a structural and physiological component of neuro-cardiology that is beginning to be appreciated. Heart disease, such as myocardial infarction, as well as several cardiomyopathies, results in abnormalities in synaptic transmission and reception. In the failing heart, the myocardial response to synaptic signalling is desensitized, resulting in a pathological, non-compensatory upregulation of synaptic transmission (Gardner et al., 2016). Following ischaemic injury, CM loss results in mass denervation of synaptic nerves. Understanding the mechanisms that regulate homeostatic innervation and interconnectivity between the two cell types lends a new perspective on pathological denervation that takes place during cardiac disease. Moreover, identifying how myocardial injury can impair nerve viability may reveal new

molecular underpinnings for cardiac dysautonomia. Identifying the mechanisms of the bidirectional signalling between CMs and SNs will be applicable for future investigations of cardiac pathologies and the development of targeted therapeutics.

Recent studies have led to a deeper recognition of sympathetic physiology, where the role of SNs extends beyond the fight-or-flight response and towards directly influencing intracellular signalling. In contrast, the level of parasympathetic innervation of the ventricular myocardium remains poorly defined. Thus, future studies are needed to map parasympathetic innervation of the ventricles and determine whether similar communication between CMs and other cell types takes place. Collectively, these findings identify a mechanistic basis for neurogenic regulation of the heart and provide novel insights into the physiology of cardiac homeostasis and disease.

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

## Additional information

### Competing interests

None.

### Author contributions

Both authors have read and approved the final version of this manuscript and agree to be accountable for all aspects of the work in ensuring that questions related to the accuracy or integrity of any part of the work are appropriately investigated and resolved. All persons designated as authors qualify for authorship, and all those who qualify for authorship are listed.

### Funding

This work was supported by an AHA Predoctoral Training Award 829586 (R.J.S.), an NIH/NHLBI grant HL155617-01A1 (A.I.M.) and a DOD grant PR210594 (A.I.M.).

### Keywords

cardiomyocytes, nerve growth factor, neuro-cardiac junction, sympathetic nerves

### Supporting information

Additional supporting information can be found online in the Supporting Information section at the end of the HTML view of the article. Supporting information files available:

**Peer Review History**

