## [Peer Review History · The Journal of Physiology]

Bridging the Communication Gap: Cardiomyocytes Reciprocate Sympathetic Nerve Signaling

Rebecca J. Salamon and Ahmed I. Mahmoud
DOI: 10.1113/JP283173

Corresponding author(s): Ahmed Mahmoud (aimahmoud@wisc.edu)

Review Timeline:

Submission Date:

29-Apr-2022

Accepted:

06-May-2022

Senior Editor: Bjorn Knollmann

Reviewing Editor: Jian Shi

Transaction Report:

Dear Dr Mahmoud,

Re: JP-P-2022-283173 "Bridging the Communication Gap: Cardiomyocytes Reciprocate Sympathetic Nerve Signaling" by Rebecca J. Salamon
Ahmed I. Mahmoud

I am pleased to tell you that your invited Perspective article has been accepted for publication in The Journal of Physiology.

NEW POLICY: In order to improve the transparency of its peer review process The Journal of Physiology publishes online as supporting information the peer review history of all articles accepted for publication. Readers will have access to decision letters, including all Editors' comments and referee reports, for each version of the manuscript and any author responses to peer review comments. Referees can decide whether or not they wish to be named on the peer review history document.

The last Word version of the paper submitted will be used by the Production Editors to prepare your proof. When this is ready you will receive an email containing a link to Wiley's Online Proofing System. The proof should be checked and corrected as quickly as possible.

All queries at proof stage should be sent to tjp@wiley.com

Thank you very much for your contribution to The Journal of Physiology.

Yours sincerely,

Bjorn Knollmann
Senior Editor
The Journal of Physiology

EDITOR COMMENTS:

Reviewing Editor:

Thank you for the nice and insightful perspective.

Senior Editor:

Excellent perspective, I concur with the reviewing editor

Reviewer Comments:

Referee #1:

The manuscript by Salamon and Mahmoud represents an accurate perspective point on the paper by Dokshokova et al., factually correct and highlights the topical findings of the manuscript, which are discussed in the broader context of physiologic and pathologic implications.